**Data Availability Statement:** All relevant data are within the manuscript and its Supporting Information files.

# "It's disappointing and it's pretty frustrating, because it feels like it's something that will never go away." A qualitative study exploring individuals' beliefs and experiences of Achilles tendinopathy

Jeffrey Turner[1☯], Peter Malliaras[2‡], Jimmy Goulis[2‡], Seán Mc Auliffe🆔[3☯]*

1 56th Medical Group, Luke Air Force Base, United States Air Force, Phoenix, AZ, United States of America, 2 Department of Physiotherapy, School of Primary and Allied Health, Monash University, Melbourne, Australia, 3 Department of Physical Therapy and Rehabilitation Sciences, College of Health Sciences, Qatar University, Doha, Qatar

☯ These authors contributed equally to this work.
‡ These authors also contributed equally to this work.
* sean@qu.edu.qa

## Abstract

### Background

Achilles tendinopathy (AT) is a common and often persistent musculoskeletal disorder affecting both athletic and non-athletic populations. Despite the relatively high incidence there is little insight into the impact and perceptions of tendinopathy from the individual's perspective. Increased awareness of the impact and perceptions around individuals' experiences with Achilles tendinopathy may provide crucial insights for the management of what is often a complex, persistent, and disabling MSK disorder.

### Purpose

To qualitatively explore the lived experiences of individuals with AT.

### Design

A qualitative, interpretive description design was performed using semi-structured telephone interviews.

### Methods

Semi-structured interviews were conducted on 15 participants (8 male and 7 female) with AT. Thematic analysis was performed using the guidelines laid out by Braun and Clarke. The study has been reported in accordance with the consolidated criteria for reporting qualitative research (COREQ) checklist.

**Funding:** The Open Access publication of this article was funded by the Qatar National Library. The funder had no role in study design, data collection and analysis, decision to publish, or preparation of the manuscript.

**Competing interests:** The authors have no financial or non-financial competing interests to declare.

## Results

Four main themes were identified from the data: 1) beliefs and perceptions surrounding AT: *"If I'm over training or something, I don't really know"*, 2) the biopsychosocial impact of AT: "*I think it restricts me in a lot of things that I would be able to do"*, 3) individuals' experiences with the management process: "*You want it to happen now. You're doing all this stuff and it's just very slow progress"*, and 4) future prognosis and outlook in individuals with AT: "*I see myself better"*.

## Conclusions

This study offers a unique insight into the profound impact and consequences of Achilles tendinopathy in a mixed sample of both athletic and non-athletic individuals. The findings of this study have important clinical implications. Specifically, it highlights the need for clinicians to recognize and adopt treatment approaches to embrace a more biopsychosocial approach for the management of tendinopathy.

## Introduction

Achilles tendinopathy (AT) is a highly prevalent, disabling and often-persistent musculoskeletal (MSK) condition, commonly affecting both athletic and non-athletic populations. In athletic populations the occurrence of AT is highest among individuals who participate in middle- and long-distance running, orienteering, track and field, tennis, badminton, volleyball, and soccer [1]. Over the past decade there has been an increased awareness of AT in non-athletic populations, challenging a prevailing belief that AT only occurs in athletic populations. Emerging evidence suggests that 2 of 3 patients with AT are not active in sports, with a suggested prevalence of 2.01 per 1,000 found in a study of Dutch general practice [2]. Unfortunately, AT is often associated with high recurrence rates and persistence of symptoms, resulting in a significant impact on individual function and disability [3]. Furthermore, treatment outcomes in AT are highly variable. In a 5-year follow-up of people treated conservatively, >60% experience continued symptoms, >40% develop contralateral symptoms, while approximately 50% of individuals rate their satisfaction as moderate-poor following treatment [3, 4].

To date, attempts to explain the pathoaetiology of tendinopathy have utilised theoretical models and paradigms, predominantly biomedical models, by focusing on the contribution of structural factors or changes in the physical properties of the tendon. This has led to biomedical or structural focused interventions in this population. Although structural factors may indeed play an important role in tendinopathy, persistent MSK pain such as AT may involve a complex interaction of numerous interlinking variables and thus should be viewed from a biopsychosocial approach [5, 6]. The role of psychosocial factors in AT has been largely neglected in the management of tendinopathy. Psychosocial factors such as mood, vigilance, self-efficacy, and personality factors have been reported to influence a person's pain experience [7, 8] and have been identified as important prognostic indicators for treatment outcomes across a range of MSK disorders [9–13].

Qualitative research provides an opportunity to address the lack of appropriate research into the lived experience of those with AT. Qualitative methods offer a platform to best understand complex psychosocial processes by capturing essential aspects of a phenomenon or

health problem from the perspective of study participants and to uncover beliefs, values, and motivations that underlie that individual's health behaviours [14]. The importance of qualitative research methods on the exploration of MSK disorders is extremely relevant in the context of moving towards a patient-centred paradigm in healthcare [15].

Thus far, only one qualitative study in AT exists, which explored the perceptions and experiences of eight people with AT [16]. The study by Mc Auliffe et al. [16] outlined the significant psychosocial impacts of AT and highlighted the need for additional research, including the area of the non-athletic population. Increased awareness of psychosocial factors and participants' experiences of AT may provide crucial insights for the management of what is often a complex, persistent, and disabling MSK disorder. Therefore, the aim of this study is to explore the lived experiences of individuals with AT.

## Methods

### Design and setting

A convenience sample of potential participants were identified in Melbourne, Australia via running clubs and from patients attending a large private practice specializing in the management of MSK disorders. Individuals meeting the specified inclusion criteria were invited to participate in the study. Potential participants were contacted via e-mail to invite participation in the study. Written consent was obtained prior to interview. In instances where invitees did not respond or refused to participate, we continued to invite additional suitable participates from our sampling frame. Sampling continued until thematic saturation was achieved, with two co-coders agreeing that no new themes were emerging.

### Inclusion and exclusion criteria

Inclusion criteria for AT participants comprised of the following: participants could be athletic or non-athletic, 18–75 years old; localized Achilles pain for >3 months duration of symptoms; gradual onset of Achilles tendon pain (subjective reporting); pain aggravated during or after weight-bearing activity; and evidence of Achilles thickening, hypoechoic regions, and/or Doppler signal on ultrasound imaging. Ultrasound imaging was performed by one experienced physiotherapist (PM) trained in ultrasound imaging and has imaged over 1000 Achilles tendons. Individuals with both mid portion and insertional tendinopathy were eligible to participate in the study.

The exclusion criteria included: under the age of 18, non-English speakers, any concurrent injuries to the foot, ankle, knee, and/or hip on the same side of the AT, current low back pain, and/or history of inflammatory arthropathy (e.g. rheumatoid arthritis).

### Study design

A qualitative interpretive description design was chosen as the methodological approach, replicating an earlier qualitative study [16]. Interpretive description is a non-categorical methodological approach created to allow healthcare practitioners to explore clinically occurring phenomena within a conducive framework [17]. This approach facilitates the exploration of complex, experiential phenomena and provides direction in the creation of an interpretative account using techniques of reflective, critical examination [18, 19]. Due to the individual experiences of AT, semi-structured interviews were employed. The questioning route followed a similar template used in the study by Mc Auliffe et al. [16; S1 Appendix] which was generated based on a literature review of relevant research [19].

## Data collection

Semi-structured telephone interviews were performed by a member of the research team (JG) who was unknown to the participants and was guided by a flexible questioning route. JG is a Physiotherapist with extensive clinical experience. The questioning route explored: participant's history of AT, perceived cause of AT, experience in managing AT, perspective on future prognosis, as well as preferred sources and format for obtaining information on their MSK pain (S1 Appendix). Prior to conducting the interviews, the interviewer (JG) undertook several practice interviews with feedback provided by a member of the investigation team (SMA) who has previous experience in conducting qualitative interview methods. Interviews lasted for approximately 30–60 minutes. Interviews were recorded using a digital voice recorder. During the interviews the researcher took notes, as needed, and statements of relevance and contextual field notes were written verbatim. At the conclusion of each interview, the interviewer debriefed the participant on the main content of the interview, and time was permitted for any additional commentary to facilitate the emergence of new unanticipated information [20].

## Data analysis

A thematic analysis approach was utilised according to Braun and Clarke [21]. Firstly, interviews were transcribed verbatim. Three transcripts were randomly selected, and initial inductive codes were formed individually by two authors (SMA and JT). The initial code lists were then amalgamated, and a comprehensive code list was finalised. The final code list was developed using the codes most representative of the dataset. The finalised code list was then applied to all transcripts by the study authors (SMA and JT). Coded data was categorised, and themes were identified through a process of repetitive interpretation, synthesising and theorising [18]. Transcripts were then re-read several times and the selected themes were finalised based on consensus discussion between study authors (SMA, JT, JG and PM). The consolidated criteria for reporting qualitative research (COREQ) checklist provided guidance during the reporting of this study [22]. To ensure the rigor of our data collection and analysis, widely accepted strategies of trustworthiness in qualitative research were adopted, including credibility, transferability, confirmability, and reflexivity [23]. Credibility was established by having two separate researchers reading, coding, and analysing the transcripts and by using quotes throughout the results to ensure the themes were rooted in the data. The current study's transferability was addressed through thick description of our participants' demographic, subjective, and objective data (Table 1 and Table 2). Audit trails describe the research steps taken from start to finish and are a foundational approach to establishing the dependability and confirmability of qualitative research findings, this study's audit trail is attached as a supplementary appendix (S1 Appendix and S2 Appendix) [23, 24]. Lastly, reflexivity was established by maintaining a constant, open, and reflective dialogue between the authors during the coding and thematic analysis from each interview to the final stages of the study [23].

In the results, sub-categories are presented for each of the four main themes, supported by quotes indexed by the participant identification number; for example, (P1). Consistent with a qualitative approach, our objective was not to quantify participant responses. However, to provide the readers with an indication of the frequency of agreement of each theme, we have used the terms "all" (15 participants); "nearly all" ($>$ 11 participants); "majority" (8–11 participants); "several" (4–7 participants); and "a few" ($<$ 4 participants).

## Ethical approval

Local research ethics board approval was obtained for this study. Ethical approval for the study was granted by Monash University Human Ethics Committee (Ethics Number: 10006).

**Table 1. Individual demographics, subjective, and objective data.**

| Subject | Age (Years) | Sex | Affected side | Location of symptoms | VISA-A | Symptom duration (months) | Previous episodes | Pain at rest (VAS) | Pain during activity (VAS) | Years of running | Running per week (distance) |
|---|---|---|---|---|---|---|---|---|---|---|---|
| 1 | 60 | M | Unilateral | Mid-portion | 78 | 4 | Yes | 0/10 | 4/10 | 5 | 30–50km |
| 2 | 38 | M | Unilateral | Insertional | 68 | 12 | No | 2/10 | 6/10 | 4 | 20–30km |
| 3 | 30 | M | Unilateral | Mid-portion | 68 | 7 | Yes | 2/10 | 4/10 | 9 | 50–60km |
| 4 | 30 | F | Unilateral | Insertional | 85 | 5 | Yes | 0/10 | 2/10 | 10 | 60km |
| 5 | 44 | F | Unilateral | Insertional | 66 | 3 | Yes | 0/10 | 1/10 | 10 | 20-30km |
| 6 | 36 | M | Unilateral | Insertional | 79 | 6 | Yes | 0/10 | 4/10 | 20 | 30–40km |
| 7 | 29 | F | Unilateral | Mid-portion | 92 | 9 | No | 2/10 | 5/10 | 2 | 20–30km |
| 8 | 49 | M | Unilateral | Insertional | 59 | 4 | Yes | 3/10 | 3/10 | > 10 | 50km |
| 9 | 26 | F | Bilateral | Insertional | 63 | 3 | Yes | 1/10 | 5/10 | > 10 | 60km |
| 10 | 35 | F | Unilateral | Insertional | 19 | 3 | No | 2/10 | 5/10 | 5 | 30km |
| 11 | 55 | M | Bilateral | Insertional | 54 | 5 | Yes | 0/10 | 4/10 | NA | NA |
| 12 | 66 | F | Unilateral | Insertional | 44 | 16 | No | 2/10 | 4/10 | NA | NA |
| 13 | 59 | M | Unilateral | Insertional | 27 | 24 | Yes | 5/10 | 9/10 | NA | NA |
| 14 | 72 | F | Bilateral | Mid-portion | 33 | 10 | Yes | 1/10 | 5/10 | NA | NA |
| 15 | 49 | M | Bilateral | Insertional | 46 | 96 | Yes | 0/10 | 3/10 | NA | NA |

M: Male; F: Female; NA: Not applicable; km: kilometres; VAS; Visual Analogue Scale; VISA-A; Victorian Institute of Sport Assessment-Achilles Questionnaire.

# Results

## Participants

Fifteen participants (8 male and 7 female) with AT were invited to participate in the telephone interviews. Table 1 details the specific demographics, objective, and subjective data of the included participants. Table 2 details the mean characteristics across the participants. Subjective and objective data collected included: Victorian Institute of Sport-Achilles questionnaire, AT pain at rest, AT pain during activity, duration of symptoms, running experience, and average weekly running distance. Elements of the audit trail are detailed in S2 Appendix.

## Key themes

Four main themes were identified from the data: 1) beliefs and perceptions surrounding AT, 2) the biopsychosocial impact of AT, 3) individuals' experiences with the management process, and 4) future prognosis and outlook in individuals with AT. Table 3 details the sub-categories constituting each theme.

**Table 2. Demographic, subjective, and objective data (mean).**

| | Mean, (participants) |
|---|---|
| Age, total, year | 45.2, (n = 15) |
| Age, runners, year | 37.7, (n = 10) |
| Age, non-runners, year | 60.2, (n = 5) |
| VISA-A | 58.7, (n = 15) |
| Duration of symptoms, months | 8.0, (n = 15) |
| Pain at rest, VAS | 1.3, (n = 15) |
| Pain during activity, VAS | 4.0, (n = 15) |

VAS: Visual Analogue Scale; VISA-A; Victorian Institute of Sport Assessment-Achilles Questionnaire; n: sample.

**Table 3. Identified main themes and sub-themes.**

| Beliefs and perceptions surrounding AT | The biopsychosocial impact of AT | Individuals' experiences with the management process in AT | Future prognosis and outlook in individuals with AT |
|---|---|---|---|
| Beliefs surrounding causation | Impact on daily routine | Motivations to seek treatment | Positive prognostic outlook |
| Perceptions regarding non-resolution in AT | Impact on running activities | Experience with passive treatment | Negative prognostic outlook |
| Frustration with providers/education | Psychological impact | Experience with active treatment | Positive self-efficacy |
| Maladaptive beliefs and avoidance behaviors | | Motivational barriers | Negative self-efficacy |

## Theme 1: Beliefs and perceptions surrounding AT–*"If I'm overtraining or something, I don't really know."*

**Beliefs surrounding causation.**   Nearly all of the participants (12/15) believed that over-training and/or overuse was a primary cause of their AT. Perceived lack of recovery or lack of time devoted to injury prevention were also common themes interwoven into their beliefs regarding overtraining.

*"So, I feel that I probably over trained. Not so much leading up to the run, it was more I didn't recover and allow myself time to recover afterwards and I just pushed it a little bit too far."* (P10)

Several participants (4/15) held strong beliefs that lack of overall fitness was the primary driver for their development of AT.

*"I just assume that I've become slightly unfit and that I always had tight muscles in my legs and it's kind of a consequence of decades of not really exercising."* (P15)

**Perceptions regarding non-resolution in AT.**   Several participants (7/15) described having lack of knowledge or being confused as to why their condition had not resolved.

*"I feel like I should know more, but I don't."* (P4)

*"I don't actually know what's going on. When I feel the pain, I mean I feel it in the base of my Achilles, but I don't know what's going on."* (P2)

**Frustration with healthcare providers.**   Frustrations and/or dissatisfaction with health-care providers and the education they received was a common theme and was reported by several participants (5/15). Often frustration stemmed from confusion derived from conflicting information from various healthcare practitioners (HCP's), lack of HCP time spent explaining the condition and required treatment, and an overall sense of HCP's not listening to each individual's complaints and questions.

*"Sometimes when you talk to your doctor, or the specialist, it's very limited time, and they don't have the time to explain it properly, and they speak in technical terms. I thought the physio spent a bit more time with you to talk to you about it."* (P13)

*"Everyone has their sort of different opinions, so it's not, I don't know, not always consistent information at the same time, as well."* (P4)

**Maladaptive beliefs and avoidance behaviours.**   Modified or ceased physical activity and exercise was a common theme throughout the interviews. Nearly all participants (12/15) reported having significantly reduced or completely ceased certain physical activities due to fear of further injury, damage, and/or pain.

*"I often just pull out earlier then. . . I never let it get that bad, if you know what I mean? I don't really go in as hard. I've got that kind of doubt niggling in the back of my mind about it. That I need to protect it, rather than let it get too bad. So I'm not someone who would take it that far to the edge. I think that's probably more of it, it just hinders me from going further or harder, or any of those things really."* (P3)

Several participants (7/15) expressed fear of tendon rupture as the reason they had stopped and/or avoided certain physical activities and hobbies. A few participants (4/15) expressed perceptions of vivid pathological damage to their tendon when asked why they thought they had not gotten better initially.

*"I'm quite certain that if I played table tennis, it would definitely be worse. I'm not even attempting that, because I'm just scared that I might rupture a tendon."* (P14)

*"If I'm overtraining or something, I don't really know, perhaps the rubbing together of the tendons causing mini fractures?"* (P7)

## Theme 2: The biopsychosocial impact of AT–*"I think it restricts me in a lot of things that I would be able to do"*

**Impact on daily routine.**   All participants (15/15) reported that their daily routines and activities were affected by their AT. This would often include social activities: e.g. going on a walk at lunch with co-workers and playing with their children.

*"I think it restricts me in a lot of things that I would be able to do. I don't think I can go out and kick the footy with my son, or. . . You know, I manage to. . . in pain, to go for a walk with the dog in the evening, if you know what I mean?"* (P11)

The majority of participants (10/15) had sedentary jobs and thus reported having pain while at work, but not reported to be a hindrance to work productivity.

*"Not a real impact on my work. I guess I've got a reasonably sedentary job."* (P8)

**Impact on running.**   Nearly all the participants (12/15) reported having reduced or completely stopped running due to their AT. A warming up effect was often cited, where they'd initially feel stiff and painful during running and the tendon would "loosen up" after a few minutes.

*"It just means altering, I guess, my training regime that I'm used to, to kind of fit in with the injury. So, when there's the running aspect in that, I just don't even bother trying anymore."* (P10)

**Psychological impact.**   A majority of participants (11/15) described frustration and/or annoyance with their condition and its limitations on activity. Several participants (7/15)

strongly identified themselves with being runners and consequently demonstrated a loss of this identity, secondary to an inability to perform at their prior levels.

> *"It's disappointing and it's pretty frustrating, really, because it feels like it's something that will never go away, but yeah, it's just very frustrating is probably the biggest thing, really."* (P4)

> *"Well, I think it's just like there's things that I enjoy doing and if I can't do them, now I get a bit frustrated and it's part of what makes me happy and makes me satisfied with things. Yeah, I think it's part of those basic sort of. . . You know, you do a nice, long run and you feel quite good after it. I'm not having that experience. I think that satisfaction, the challenge, and all that sort of stuff, I'm just not being able to do and expose myself to and I kind of struggle to find that in other modes when I'm not running."* (P8)

## Theme 3: Individuals' experiences with the management process in AT–*"You want it to happen now. You're doing all this stuff and it's just very slow progress"*

**Motivations to seek treatment.**    A majority of participants (11/15) were motivated to seek treatment by their pain and fear of disability, worsening condition, or further loss of physical activity.

> *"And there are still things I want to do in the future, like with running and more marathons, and even doing some ultras and stuff like that. So, I don't have a choice but to keep it strong and keep doing those exercises."* (P10)

**Experience with passive treatment.**    Nearly all of the participants (14/15) were prescribed or sought passive treatment for management of their AT. Passive treatments (e.g. massage, dry needling, ultrasound, etc.) were the most cited form of preferred and reported most effective treatment type. Massage treatment being the most common form of passive modality prescribed or sought by participants.

> *"I've had a lot of massage over the years. I've had dry needling in my calves because my calves and hamstrings are tight, and this may be contributing to my symptoms."* (P4)

> *"So, acupuncture tends to. . . I respond really well to that and pretty quickly as well. Obviously, massage, anything to loosen up my calf, really. So, massage work or acupuncture on my calves."* (P9)

**Experience with active treatment.**    Several of the participants (6/15) believed strength training was the most effective treatment in AT.

> *"I'm not sure that the exercise alone, without the shock wave, would've been effective."* (P14)

Nearly all of the participants (13/15) were prescribed some form of strength training by HCPs. The most frequently reported type of prescribed strength training was body weight calf raises/heel lifts. Dosages were quite variable and nearly all participants (13/15) reported difficulty with maintaining adherence with their prescribed exercises. A majority of participants (10/15) blamed themselves for the cause of their AT.

> *"I'm the only one to blame for it being the way it is. Yeah, I mean it's certainly in my control. I can't blame anyone else for it being the way it is and I've. . . It's my decision to do or not do*

*my exercises and things like that. I wish there was a magic pill that I could take to resolve it."*
(P2)

*"I failed to really maintain it properly. I should have done more of those exercises prescribed"*
(P11)

**Motivational barriers.** A few participants (4/15) reported lack of motivation to seek treatment. Common barriers noted were feelings of lack of control over their condition, perceptions that rehabilitation exercise was tedious and not enjoyable, and frustration with the slow progress of rehabilitation.

*"I failed to go to the gym this morning, because I was feeling fed up with myself and so I'm not really in control of any of these things."* (P15)

## Theme 4: Future prognosis and outlook in individuals with AT–"*I see myself better*"

**Positive prognostic outlook.** A majority of participants (10/15) expressed optimism regarding the future of their condition and return to prior levels of activity. The self-identified need for compliance with prescribed treatment and exercises was identified as a driver of future improvement.

*"I see myself better. I think I feel like I'm very much in the last stages of this problem. I see myself better and I think of one thing I've progressed with is in learning ways to ongoingly [sic] prevent this sort of injury."* (P8)

**Negative prognostic outlook.** Several participants (5/15) expressed a negative outlook on their future prognosis and ability to return to prior levels of activity. Citing chronicity, pathology, and genetic disposition as reasons for poor prognosis.

*"It can be relieved, but there's nothing you can do about flat feet. I'm born like that. That's how life is. I'm not gonna [sic] play tennis again. I'm not sure that I'll be able to play table tennis."* (P14)

**Positive self-efficacy.** A majority of participants (8/15) reflected confidence in their ability to exert control over their condition.

*"It's something that's not going to go away, and if I don't keep up doing the exercises, it's just going to weaken. I'm always mindful to keep it strong and keep it going well, I guess."* (P5)

**Negative self-efficacy.** Several participants (7/15) reflected a lack of confidence in their ability to exert control over their condition.

*"Yeah, I kind of almost resigned myself that it's going to be a long process."* (P8)

*"I no longer feel like I'm in control of it now."* (P9)

## Discussion

The current study sought to explore the lived experiences of individuals with AT and expand the findings of Mc Auliffe et al. [16] to a larger sample which included non-athletic

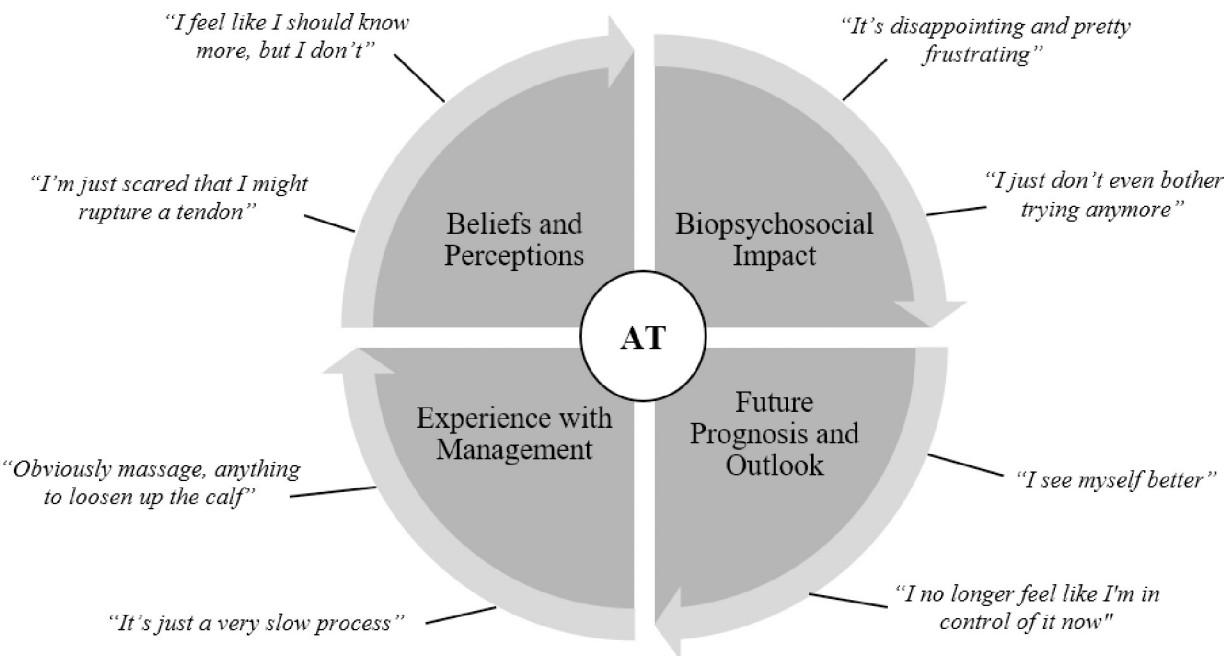

**Fig 1. Main themes and representative participant quotes.** Achilles tendinopathy (AT).

participants. Specifically, the results of this study revealed four main themes: 1) beliefs and perceptions surrounding AT, 2) the biopsychosocial impact of AT, 3) individuals' experiences with the management process in AT, and 4) future prognosis and outlook in individuals with AT (Fig 1). The findings provide unique and important insights into the individuals' perspectives of AT, which may provide important implications for clinical practice.

## Psychosocial impact of AT

A strong theme to emerge throughout the participant interviews was the significant psychosocial impact of AT. In particular, frustration was a common and consistent sub-theme. Frustration is recognised as a significant negative emotion and has been associated with other pain-related emotions such as fear, anxiety and anger in individuals experiencing chronic MSK pain [25, 26]. Participants in the current study indicated high levels of frustration that were associated with the persistence of symptoms and reported disruption of daily activities as a result of AT. The implications of such findings are important, they indicate that MSK disorders such as AT does not exclusively affect physical capabilities but also the contextual and psychological domains of an individual's well-being and quality of life which the HCP may need to consider. The significant psychosocial impact, particularly in terms of participation in daily life and valued activities is comparable to the previous study by Mc Auliffe et al [16].

Participants in the present study also reported frustration with HCP; specifically, participants cited frustration derived from receiving conflicting information, lack of HCP time spent explaining the condition and required treatment, and an overall sense of HCPs not listening to their individual complaints and questions. These findings are similar to Dow et al. [25] who investigated the potential influence of frustration on the chronic pain experience and relationships with HCPs. Similar to the findings of the current study, Dow et al. [25] also found that people reported entering medical consultations with a range of unresolved frustrations relating to poor communication, lack of a diagnosis, and dissatisfaction with the treatment options

available. In recent years, there has been a growing interest by researchers to understand this patient-therapist relationship and its effects on physical rehabilitation and treatment outcomes. A systematic review by Hall et al. [27] investigating the role of the therapeutic alliance in physical rehabilitation concluded that enhanced patient-therapist relationships had a positive effect on treatment outcomes. However, there is little consensus on appropriate educational strategies to best improve patient outcomes in persistent MSK pain, stressing the need for further research in tendinopathy and other MSK disorders [28, 29].

## Exercise and tendinopathy: Room for improvement?

Current clinical practice guidelines and systematic reviews recommend exercise or loading based therapies as the first-line treatment for AT [30, 31, 32]. The present study found that despite most participants reporting that they had received some form of strength training or loading intervention as part of the treatment process, only 6 out of 15 participants believed that strength training exercise was the most effective form of treatment for their AT. In fact, passive treatments (e.g. massage, dry needling, ultrasound, etc.) were the most cited form of preferred and perceived effective treatment. One potential reason for the lack of belief in efficacy of exercise or strengthening based interventions identified in this study may relate to the impact of fear avoidance beliefs as a result of the pain associated with AT. The presence of fear avoidance beliefs in AT was a common theme found in this study, consistent with findings from Mc Auliffe et al. [16]. Participants identified that they had stopped or reduced normal daily activity (e.g. walking their dogs, playing with their children) and/or recreational exercise (e.g. running, tennis) secondary to fears of worsened pain, further injury, or tendon rupture. Such associated behaviours are referred to as fear avoidance beliefs, a concept first proposed in the Fear Avoidance Model (FAM) of MSK pain [8, 33]. According to this theory, some individuals consider a painful stimulus as negative and avoid or postpone the event that is considered painful [34]. Additionally, the psychological consequences of AT may lead to cognitive appraisal resulting in the generation and adoption of a compensatory behaviour to protect or avoid activities that they interpret as harmful for their tendon [35]. Research in persistent MSK conditions, including AT, have highlighted the negative influence of fear avoidance beliefs on recovery time and clinical outcome [9, 10, 28, 36]. Ongoing fear, catastrophizing, and/or anxiety associating the AT with an adaptive behaviour may lead to further deconditioning or atrophy of the MSK system [37], which may contribute to the persistence of symptoms and avoidance of physical activity demonstrated in our study [4].

Another potential explanation for participants' lack of belief in exercise efficacy may relate to a lack of motivation to engage in exercise in addition to several other barriers including: feelings of lack of control over their condition, perceptions that rehabilitation exercise was tedious and not enjoyable, and frustration with the slow progress of rehabilitation–some of which were cited in the current study. Finally, the lack of belief and minimal adherence in active exercise approaches in AT may also be attributed to the role of the HCP. Potentially, the provision of inadequate education or awareness on the importance of active approaches in favour of passive treatment modalities may exacerbate the lack of belief in active treatment approaches. A recent systematic review [38] investigated this topic by exploring if physical therapists follow evidence-based guidelines when managing MSK conditions. Worryingly, the review indicated that many physical therapists do not follow evidence-based guidelines when managing MSK conditions, and this may explain some of the findings in the current study. Whether the mismatch is 1) the communication and education between patient and provider, or 2) patients' expectations or providers' treatment approaches; the results of this study highlight the potential need to address the overall management approach in AT. Specifically, there

is a need to align management strategies to the clinical practice guidelines for MSK pain published by Lin et al [39], which included: providing patients with education/information about their condition, providing management addressing exercise, and applying manual therapy only as an adjunct to other evidence-based interventions.

## Adopting the biopsychosocial model: A way forward

In recent years, there has been a growing body of evidence recommending specific approaches to facilitate adoption of the biopsychosocial model of health into routine clinical practice [40, 41]. Specifically, clinicians have been advised to 1) adopt open and reflective questioning surrounding a patient's experiences, beliefs, biopsychosocial factors, and expectations; 2) set shared goals and discover the patient's related concerns and limitations; 3) educate beyond words using active learning approaches to facilitate behaviour change; and 4) coach towards self-management using support through understanding, reassuring, and empowerment. Based on the findings of the current study these approaches may have particular importance in the management of AT (Fig 2). However, to date there has been a distinct lack of research investigating the use of such approaches in tendon disorders. Similar issues have been highlighted in other MSK disorders where there has been minimal adoption and utilization of such models in clinical practice [42]. In fact, a systematic review by Synnott et al. [43] identified that physical therapists only partially recognized psychosocial factors' role in pain, questioned the relevance for screening these factors, and reported feeling undertrained and ill-equipped in addressing them. This highlights the need for further research on appropriate implementation of psychosocial factors in routine clinical practice in MSK pain including tendinopathy.

## Limitations and strengths

Participants were primarily recruited from one large multispecialty orthopaedic and sports rehabilitation clinic; this could limit the diversity of the participants included in the paper. Participants were not sampled based on ethnicity and future work is needed to explore the role of culture in understanding the consequences of AT. The majority of participants included in this study had insertional AT (11/15). In addition, only one author (physical therapist) conducted the interviews and transcribed the data. There is the potential that this could infuse intellectual bias and result in "leading the witness". To mitigate this, all authors were involved in the interview route design and structure and all efforts were put in place to limit the extent of questions/responses outside of the interview route, helping to establish the reflexivity of the study. The present study expanded on similar themes found in Mc Auliffe et al. [16] in a different geographical location and included non-athletic participants, building the credibility and confirmability of the study. Finally, to further enhance the trustworthiness and transparency of this study we've attached, as supplementary files, our interview questions and audit trail (S1 Appendix and S2 Appendix).

## Conclusion

Although our understanding about tendinopathy continues to evolve, the prevailing methodology to date in tendinopathy has been dominated by objective, quantitative approaches over qualitative exploration. Results of the current study add to an emerging body of evidence highlighting the substantial fears, frustrations and impact on quality of life and daily functioning in individuals with tendinopathy. The exclusive focus on quantitative methodologies to the expense of qualitative research approaches negates the concept of providing patient-centred care for MSK disorders, as such approaches may fail to incorporate individual patient's preferences, needs, and values [44]. Results of this study emphasize the need to address psychosocial

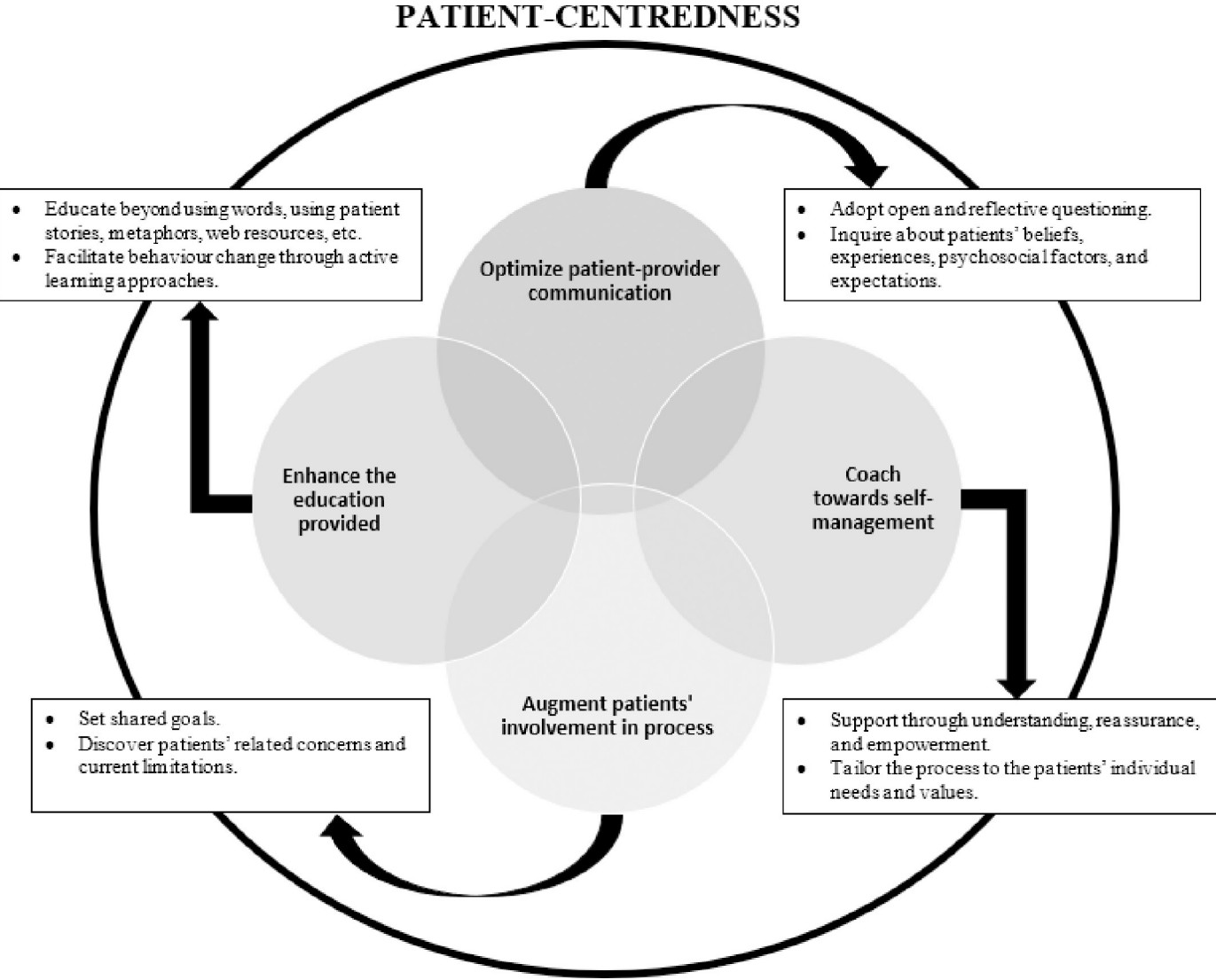

**Fig 2. Proposed patient-centredness model.**

factors in AT, a point that was highlighted in the recent international scientific tendinopathy symposium consensus paper which recommended including psychological factors as a core health-related domain for the assessment and treatment of tendinopathy [45]. Such insights may have profound implications for informing appropriate clinical practice and underscores the potential for a biopsychosocial approach in the management of AT.

## Supporting information

**S1 Appendix. Interview question route.**
(DOCX)

**S2 Appendix. Audit trail.**
(DOCX)

**S1 File. COREQ (COnsolidated criteria for REporting Qualitative research) checklist.** (PDF)

## Acknowledgments

The authors thank the participants of this study. The views expressed here are those of the authors and do not reflect the official policy or position of Luke Air Force Base, the Department of the Air Force, the Department of Defense, or the U.S. government.

## Author Contributions

**Conceptualization:** Peter Malliaras, Jimmy Goulis, Seán Mc Auliffe.

**Data curation:** Peter Malliaras, Jimmy Goulis.

**Formal analysis:** Jeffrey Turner, Peter Malliaras, Jimmy Goulis, Seán Mc Auliffe.

**Investigation:** Peter Malliaras.

**Methodology:** Jeffrey Turner, Peter Malliaras, Jimmy Goulis, Seán Mc Auliffe.

**Resources:** Peter Malliaras, Jimmy Goulis, Seán Mc Auliffe.

**Software:** Jimmy Goulis.

**Supervision:** Peter Malliaras, Seán Mc Auliffe.

**Validation:** Jeffrey Turner.

**Visualization:** Jeffrey Turner, Seán Mc Auliffe.

**Writing – original draft:** Jeffrey Turner, Peter Malliaras, Jimmy Goulis, Seán Mc Auliffe.

**Writing – review & editing:** Jeffrey Turner, Peter Malliaras, Jimmy Goulis, Seán Mc Auliffe.

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
