## [Decision Letter · Decision Letter 0]

14 Apr 2020

PONE-D-19-35845

“It's disappointing and it's pretty frustrating, because it feels like it's something that will never go away.” A qualitative study exploring individuals’ beliefs and experiences of Achilles tendinopathy

PLOS ONE

Dear Dr. McAuliffe,

Thank you for submitting your manuscript to PLOS ONE. After careful consideration, we feel that it has merit but does not fully meet PLOS ONE’s publication criteria as it currently stands. Therefore, we invite you to submit a revised version of the manuscript that addresses the points raised during the review process.

ACADEMIC EDITOR: 

address each of the comments from the reviewer.clarify the issue around irrational fear of reinjury.

We would appreciate receiving your revised manuscript by May 29 2020 11:59PM. To enhance the reproducibility of your results, we recommend that if applicable you deposit your laboratory protocols in protocols.io, where a protocol can be assigned its own identifier (DOI) such that it can be cited independently in the future. For instructions see: http://journals.plos.org/plosone/s/submission-guidelines#loc-laboratory-protocols

We look forward to receiving your revised manuscript.

Kind regards,

Denis Martin, PhD

Academic Editor

PLOS ONE

Journal Requirements:

3. Please amend the manuscript submission data (via Edit Submission) to include authors: Jeffrey Turner, Peter Malliaras and Jimmy Goulis

Additional Editor Comments (if provided):

Thank you for your submission and your patience in awaiting a decision. The paper is well written and the qualitative approach adds useful knowledge to the theory and practice of managing achillies tendon injuries.

In addition to the reviewer's comments, which should all be addressed in full, I have added one further point to be addressed. This concerns the finding of fear of movement. Within the context of a biopsychosocial approach to pain, which is where this paper is set, fear of movement or kinesiophobia is usually meant to imply an irrational fear of potential damage. That is well especially well established in low back pain where injury should not be an expected outcome of movement. However, as acknowledged in the introduction, there is a sizeable rate of reinjury  in achillies injuries that belies a clear justification for calling fear of fully loading the achilles as irrational. The authors should clarify in 1.4 what exaclty is maladaptive in the beliefs and behaviour they have observed.

Reviewers' comments:

Reviewer's Responses to Questions

**Comments to the Author**

1. Is the manuscript technically sound, and do the data support the conclusions?

Reviewer #1: Yes

2. Has the statistical analysis been performed appropriately and rigorously? 

Reviewer #1: N/A

3. Have the authors made all data underlying the findings in their manuscript fully available?

Reviewer #1: No

4. Is the manuscript presented in an intelligible fashion and written in standard English?

Reviewer #1: Yes

5. Review Comments to the Author

Reviewer #1: Thank you for the opportunity to review this manuscript. The authors are to be commended on a well written and very interesting paper that has some useful insights for clinical practice. I have made a number of minor points below - the main issue is the need to provide additional information about key components of methodological rigour.

Abstract

L30 Need to put in the acronym for AT here the first time you use the full term so that the acronym later in the abstract makes sense.

L31 change disorders to disorder

L32 There is LITTLE insight?

L50 The authors talk about a two cohorts an athletic and non-athletic cohort – I think it would be clearly to say that the sample was a mixture of individuals from an athletic and non-athletic background.

Introduction.

It would be good to know what the previous study these authors have found (reference 16) and clarify how this current study seeks to extend those findings. Then in the discussion clarify if this work did build on those previous findings and how?

Methods

L130 Allow health care practitioners TO explore

The methods section would benefit from additional detail regarding the demographic and any questionnaire data (the VISA-Q and the VAS) that were collected.

The authors state they used convenience sampling. This is fine, but there seems to have been an interest in recruiting both athletes and non athletes what would the authors have done if their convenience sampling only produced athletes (or vice versa). It seems purposive sampling might have been more appropriate.

The methods requires additional information about key rigour components in qualitative methods – e.g. credibility, trustworthiness, dependability and reflexivity – defining these and discussing how these were addressed e.g. two different individuals reading the transcripts and using quotes to ensure the themes were rooted in the data. Ensuring all opinions were included including those who were going against the grain, what the authors backgrounds are regarding AT and how this may or not have influenced the work etc… This is probably the key revision that is required.

I cannot see if the data has been made available as per PLOS one requirements – apologies if I have missed this.

Results

Tables 1 and 2 would benefit from a legend covering things like what do the acronyms mean.

Figure 1 is excellent – very concise and informative

Discussion

L478 review by [38] – delete the word BY

6. PLOS authors have the option to publish the peer review history of their article (what does this mean?). If published, this will include your full peer review and any attached files.

Reviewer #1: No

---

## [Author Response · Author response to Decision Letter 0]

26 Apr 2020

We thank the reviewers for their thoughtful and thorough review of our manuscript. We have taken every comment into consideration and responded to them individually below.

EDITOR:

Within the context of a biopsychosocial approach to pain, which is where this paper is set, fear of movement or kinesiophobia is usually meant to imply an irrational fear of potential damage. That is well especially well established in low back pain where injury should not be an expected outcome of movement. However, as acknowledged in the introduction, there is a sizeable rate of reinjury in Achilles injuries that belies a clear justification for calling fear of fully loading the achilles as irrational. The authors should clarify in 1.4 what exactly is maladaptive in the beliefs and behavior they have observed.

Author’s Response:

Thank you for your comments. We have clarified 1.4 by amending the subtitle to “Fear avoidance behaviors”, to better reflect our results and discussion. L470-488 in our discussion section elaborates on the results found in 1.4 in relation to how our participants’ fear of rupture/damage detrimentally effects their participation in daily activities and recreation. Despite the sizeable reinjury risk that is present in Achilles tendinopathy, the risk of potential rupture is still quite low, prior research by Yasui et al. found 4% of patients with Achilles tendinopathy later developed an Achilles tendon rupture. In addition, exercise-based therapies are considered the front-line intervention for treatment. Furthermore, the terms fear avoidance behaviors/kinesiophobia have been used in previous studies to describe the phenomenon in Achilles tendinopathy (Mallows et al. 2017).

Yasui Y, Tonogai I, Rosenbaum AJ, Shimozono Y, Kawano H, Kennedy JG. The Risk of Achilles Tendon Rupture in the Patients with Achilles Tendinopathy: Healthcare Database Analysis in the United States. Biomed Res Int. 2017; ePub 2017 Apr 30.

Mallows A, Debenham J, Walker T, Littlewood C. Association of psychological variables and outcome in tendinopathy: a systematic review. British Journal of Sports Medicine. 2017;51(9): 743-748.

REVIEWER:

Abstract:

L30 Need to put in the acronym for AT here the first time you use the full term so that the acronym later in the abstract makes sense.

Author’s Response: 

We’ve added the acronym AT after “Achilles tendinopathy.”

L31 Change disorders to disorder

Author’s Response:

Thank you, we’ve updated this.

L32 There is LITTLE insight?

Author’s Response:

Thank you, we’ve updated this to “little insight”.

L50 The authors talk about two cohorts, an athletic and non-athletic cohort – I think it would be clearer to say that the sample was a mixture of individuals from an athletic and non-athletic background.

Author’s Response:

Good point on modifying language here to improve clarity, see update below.

Original:

“This study offers a unique insight into the profound impact and consequences of Achilles tendinopathy in a cohort of both athletic and non-athletic individuals.” 

Updated:

“This study offers a unique insight into the profound impact and consequences of Achilles tendinopathy in a mixed sample of both athletic and non-athletic individuals.”

Introduction:

It would be good to know what the previous study these authors have found (reference 16) and clarify how this current study seeks to extend those findings. Then in the discussion clarify if this work did build on those previous findings and how?

Author’s Response:

Excellent feedback. Themes from Mc Auliffe et al. were similar to those we found in our present study. Our sample was larger and included a mixed cohort of both athletic and non-athletic participants with AT, whereas Mc Auliffe et al. only had athletic participants. We’ve amended the intro, discussion, and limitations to include this which can been seen in the marked manuscript version.

Mc Auliffe S, Synott A, Casey H, Mc Creesh K, Purtill H, O'Sullivan K. Beyond the tendon: Experiences and perceptions of people with persistent Achilles tendinopathy. Musculoskeletal Science and Practice. 2017;29: 108-114.

Methods:

L130 Allow health care practitioners TO explore

Author’s Response:

Thank you, we’ve now updated this.

The methods section would benefit from additional detail regarding the demographic and any questionnaire data (the VISA-Q and the VAS) that were collected.

Author’s Response:

Thank you for the feedback. Participant data is presented in the first part of the results section, we have amended the manuscript to include additional details regarding demographic and objective data collected. See below.

Participants Section Updated:

Fifteen participants (8 male and 7 female) with AT were invited to participate in the telephone interviews. Table 1 details the specific demographics, objective, and subjective data of the included participants. Table 2 details the mean characteristics across the participants. Subjective and objective data collected included: Victorian Institute of Sport-Achilles questionnaire, AT pain at rest, AT pain during activity, duration of symptoms, running experience, and average weekly running distance. Elements of the audit trail are detailed in S2 Appendix.

The authors state they used convenience sampling. This is fine, but there seems to have been an interest in recruiting both athletes and non-athletes what would the authors have done if their convenience sampling only produced athletes (or vice versa). It seems purposive sampling might have been more appropriate.

Author’s Response:

Thank you for the feedback. The sampling was convenient in nature - we didn’t intentionally select a specific number of athletic and non-athletic participants, they conveniently presented in this manner as part of a routine private practice setting. We felt that this method of recruitment best represented the sample seen in clinical practice.

We are happy to change if the reviewers are not happy with this justification in sampling method terminology?

The methods requires additional information about key rigour components in qualitative methods – e.g. credibility, trustworthiness, dependability and reflexivity – defining these and discussing how these were addressed e.g. two different individuals reading the transcripts and using quotes to ensure the themes were rooted in the data. Ensuring all opinions were included including those who were going against the grain, what the authors backgrounds are regarding AT and how this may or not have influenced the work etc… This is probably the key revision that is required.

Author’s Response:

Thank you for your comments, we have updated the “Data Analysis” section to include the below paragraph. Please also see the “Trustworthiness techniques” section of our S2 Appendix: Audit Trail for further information.

Data Analysis Updated:

To ensure the rigor of our data collection and analysis, widely accepted strategies of trustworthiness in qualitative research were adopted, including credibility, transferability, confirmability, and reflexivity [23]. Credibility was established by having two separate researchers reading, coding, and analysing the transcripts and by using quotes throughout the results to ensure the themes were rooted in the data. The current study’s transferability was addressed through thick description of our participants’ demographic, subjective, and objective data (Table 1 and Table 2). Audit trails describe the research steps taken from start to finish and are a foundational approach to establishing the dependability and confirmability of qualitative research findings. This study’s audit trail is attached as a supplementary appendix (S1 Appendix and S2 Appendix) [23, 24]. Lastly, reflexivity was established by maintaining a constant, open, and reflective dialogue between the authors during the coding and thematic analysis from each interview to the final stages of the study [23].

I cannot see if the data has been made available as per PLOS one requirements – apologies if I have missed this.

Author’s Response:

Thank you for bringing this to our attention. We submitted the files as part of the overall submission but apologise if you were unable to access these documents. We have attached the data, supplemental appendix 1 and 2, with this re submission for your consideration.

Results:

Tables 1 and 2 would benefit from a legend covering things like what do the acronyms mean.

Author’s Response:

Thank you for bringing this to our attention. Legends have been updated to reflect all acronyms within their corresponding tables.

Discussion:

L478 review by [38] – delete the word BY

Author’s Response:

Thank you, this has been removed.

---

## [Editor Report · Decision Letter 1]

6 May 2020

“It's disappointing and it's pretty frustrating, because it feels like it's something that will never go away.” A qualitative study exploring individuals’ beliefs and experiences of Achilles tendinopathy

PONE-D-19-35845R1

Dear Dr. McAuliffe,

We are pleased to inform you that your manuscript has been judged scientifically suitable for publication and will be formally accepted for publication once it complies with all outstanding technical requirements.

With kind regards,

Denis Martin, PhD

Academic Editor

PLOS ONE

Additional Editor Comments (optional):

Thank you for your response to the comments.
---

## [Editor Report · Acceptance letter]

14 May 2020

PONE-D-19-35845R1 

“It's disappointing and it's pretty frustrating, because it feels like it's something that will never go away.” A qualitative study exploring individuals’ beliefs and experiences of Achilles tendinopathy 

Dear Dr. Mc Auliffe:

I am pleased to inform you that your manuscript has been deemed suitable for publication in PLOS ONE. Congratulations! Your manuscript is now with our production department. 

With kind regards,

on behalf of

Dr. Denis Martin 

Academic Editor

PLOS ONE